# Smartphone-Based Facial Scanning as a Viable Tool for Facially Driven Orthodontics?

**DOI:** 10.3390/s22207752

**Published:** 2022-10-12

**Authors:** Andrej Thurzo, Martin Strunga, Romana Havlínová, Katarína Reháková, Renata Urban, Jana Surovková, Veronika Kurilová

**Affiliations:** 1Department of Stomatology and Maxillofacial Surgery, Faculty of Medicine, Comenius University in Bratislava, 81250 Bratislava, Slovakia; 2Faculty of Electrical Engineering and Information Technology, Slovak University of Technology, Ilkovičova 3, 81219 Bratislava, Slovakia

**Keywords:** TrueDepth, CBCT, orthodontics, face scan, smartphone, facial diagnostics, smartphone-based sensors, facially driven orthodontics

## Abstract

The current paradigm shift in orthodontic treatment planning is based on facially driven diagnostics. This requires an affordable, convenient, and non-invasive solution for face scanning. Therefore, utilization of smartphones’ TrueDepth sensors is very tempting. TrueDepth refers to front-facing cameras with a dot projector in Apple devices that provide real-time depth data in addition to visual information. There are several applications that tout themselves as accurate solutions for 3D scanning of the face in dentistry. Their clinical accuracy has been uncertain. This study focuses on evaluating the accuracy of the Bellus3D Dental Pro app, which uses Apple’s TrueDepth sensor. The app reconstructs a virtual, high-resolution version of the face, which is available for download as a 3D object. In this paper, sixty TrueDepth scans of the face were compared to sixty corresponding facial surfaces segmented from CBCT. Difference maps were created for each pair and evaluated in specific facial regions. The results confirmed statistically significant differences in some facial regions with amplitudes greater than 3 mm, suggesting that current technology has limited applicability for clinical use. The clinical utilization of facial scanning for orthodontic evaluation, which does not require accuracy in the lip region below 3 mm, can be considered.

## 1. Introduction

The paradigm shift in orthodontic treatment planning is currently leaning towards soft tissue-driven considerations. Although 3D facial diagnosis is undoubtedly a crucial factor in treatment planning, it has been difficult to capture by the traditional means of 2D digital diagnostics. The 3D diagnostic workflow will soon be considered a routine procedure, but the affordability of high-end facial scanners is slowing this change. The ability to use the smartphone sensor introduced in 2017 in the iPhone X has given the world a new method of affordable and convenient facial scanning. It captures more than 250,000 3D data points of a face in 10 s as the patient slowly turns their head in front of the iPhone or iPad. Despite various professional dental applications proclaiming themselves as reliable and accurate solutions, the clinical accuracy has been questionable [1,2].

Dental imaging is a standard clinical procedure that is an important source of information for various purposes. Soft tissue images of the face are important records for evaluating the maxillofacial area in many fields such as orthodontics, orthognathic surgery, and facial plastic surgery, and are used for diagnostic processing, treatment planning, and outcome analysis, among others. The application of 3D scanning methods in healthcare, especially cone beam computed tomography (CBCT) imaging in the field of maxillofacial surgery and orthodontics, has expanded significantly over the last decade [3,4].

CBCT is a three-dimensional (3D) diagnostic X-ray imaging technique which provides craniofacial imaging with low distortion and higher image accuracy than conventional imaging. When the acquired data is processed in the volume, CBCT generates 3D panoramic and cephalometric pictures [5,6]. CBCT also has certain limitations. These devices are being used in dental care mainly to image the hard tissues of the orofacial structure and have a limited capacity to identify soft tissues due to the lack of contrast resolution and texture, limiting their usage for soft tissue analysis. Furthermore, the use of stabilizing tools for CBCT scanning such as chin rests or forehead restraints could deform the surface anatomy of the facial soft tissue, and it is also susceptible to a variety of artifacts, including metal and motion artifacts, which can have a negative impact on image quality [7,8]. To address the lack of soft tissue data provided by CBCT scanners, 3D facial scanners have been integrated into digital routines such as stereophotogrammetry, laser, and structured-light systems, offering a non-ionizing technique for creating a copy of the facial soft tissue with a precise portrayal of texture and static geometry in three dimensions [9] since the texture and color is significant for treatment planning in orthodontics. These complementary methods could be precise, quick, and easy to use.

Utilization of smartphones in combination with artificial intelligence is a common practice in orthodontics today. Artificial intelligence (AI) in the form of dental monitoring software uses the patient’s cell phone for regular scanning. This has advantages in the pandemic era [10], as well as in self-evaluating coaching tele-health solutions [11]. AI is also currently widely used in the diagnosis of 3D facial scans created with smartphones [12].

3D face scanning is a fast-expanding field with enormous potential in a wide range of uses, but it is still new and quite unexplored. As smartphone availability and capabilities expand, so does the potential for 3D face-scanning apps. They have a variety of uses in medicine and dentistry, such as face identification, emotion capturing, facial cosmetic planning and surgery, and maxillofacial rehabilitation.

Facial scanners can generate a 3D topography of a patient’s facial surface anatomy, which, when paired with a digital study model and a CBCT scan, creates a 3D “virtual patient” for improved diagnosis, treatment planning, and patient outcomes [13,14,15].

Recent innovations of devices such as smartphones and tablets have demonstrated that scanning is also possible using LiDAR and TrueDepth technology.

LiDAR, which stands for light detection and ranging, is a radar-like remote sensing technology. The difference is that radar detects its surroundings using radio waves, while LiDAR requires laser energy. The technology refers to a remote sensing technology that generates concentrated light beams and calculates the time taken to detect the reflections by the sensor [1,16,17,18].

For the 3D reconstruction, depth sensors are used, which have been utilized for a long time in 3D scanners, game systems (Microsoft Kinect, for example), and lately, in laptops and smartphones. These sensors are resilient in a variety of lighting conditions (day or night, with or without glare and shadows), thereby outperforming other sensor types. Sensors may be positioned at the back and at the front of the device. The depth sensors on the front of the unit have a shorter range and can identify and map hundreds of landmarks in real time; they are primarily used to detect the face and produce its 3D image. Their essential use is for biometric smartphone security (e.g., Apple Face ID), to recognize the face of the Apple device owner.

Facial recognition has improved dramatically in only a few years. As of April 2020, the best face identification algorithm has an error rate of just 0.08% compared to 4.1% for the leading algorithm in 2014, according to tests by the National Institute of Standards and Technology (NIST) [19]. To recognize faces in Apple devices (i.e., iPhone X and later), the front-facing cameras with a Dot projector provide depth data in real time along with visual information. The core technology responsible for this process is called TrueDepth. The system uses LEDs to project an irregular grid of over 30,000 infrared dots to record depth within a matter of milliseconds [20,21] and can provide a rapid, reliable, and direct method for producing 3D data [1,22,23,24].

TrueDepth (Bellus3D Dental Pro) scanning performed on an iPhone or iPad with a 3D capture system can use an affordable program for face scanning such as Bellus3D FaceApp, which has simple instructions and was created to be precise and accurate in recognizing facial landmarks [25,26,27]. It is also available for Android and iPhone devices, is compatible with the Windows 10 operating system, and it allows simple export of STL files. The first such hardware and software solution using an accessory camera for Android smartphones was created in March 2015 in Silicon Valley with aim to generate detailed 3D face scans [28,29]. Later, the company presented the FaceApp application, dedicated for iPhoneX users. The functionality of this app is the same as in the Android version, except instead of using any additional hardware devices, it used the front-facing TrueDepth camera the same way as in Apple Face ID.

Facial scanning can provide useful correlative data for many studies that would benefit from regular, noninvasive evaluations of head and neck soft-tissue morphology, as the change of body mass index is not a very representative value when the facial morphology is the merit [30,31]. Three-dimensionally printed extraoral orthodontic appliances in growing patients would benefit from regular, noninvasive and reasonably accessible facial scanning [32]. Tsolakis et al., 2022, as well, presents ideas that can be widely utilized for regular evaluations of growth or therapeutical changes of facial morphology [33].

FaceApp captures more than 250,000 3D data points on a face in 10 s while the user slowly turns their head in front of the camera. The app then reconstructs a virtual high-resolution version of the face that can be rotated, zoomed in or out, and viewed in three dimensions. Additionally, the face model can be viewed with interactive lighting, using the device’s gyro to control viewing angles. Apple just released iOS 15.4 with some improvements for the iPhone 12 and iPhone 13 when it comes to using the Face ID feature while using a mask.

Mobile phone 3D facial scanning in combination with AI algorithms incorporated in a smartphone app, for example, Face2Gene (FDNA Inc., Boston MA, USA), is currently forming a powerful tool for early diagnostics. Diseases not only manifest as internal structural and functional abnormalities, but also have facial characteristics and appearance deformities. Specific facial phenotypes are potential diagnostic markers, especially for endocrine and metabolic syndromes, genetic disorders, and facial neuromuscular diseases [34,35].

The goal of this work was to assess whether the facial scan created with TrueDepth sensors and compiled with the Bellus Dental Pro app is accurate compared to the surface of the face from the CBCT, and, in the case of inaccuracies, to determine which facial regions are incorrectly imaged and to what extent.

## 2. Materials and Methods

This paper is focused on the analysis and deviations of facial soft tissue scans between CBCT and the TrueDepth scanner using the Bellus 3D FaceApp application, respectively. Specific facial areas, attributes, and points were chosen for evaluation in these scans using programs Invivo 6 (Anatomage Inc., San Jose, CA, USA) Dental and Meshmixer™ (Autodesk^®®^, Inc., San Rafael, CA, USA).

The assessment of the accuracy of the facial scan was based on the CBCT scan, which accurately reflects the true morphology of the face. From the CBCT of the head, a portion representing the face was segmented and exported as an STL shell. The difference between the polygon resolution of the STL from the CBCT and the STL from the face scan (TrueDepth—Bellus3D Dental Pro) is obvious, as shown in Figure 1.

The methods presented can be divided into 3 parts:Collecting scans;Modification, positioning, and analysis of the scans;Data analysis and comparison.

### 2.1. Collecting Scans—Selection Criteria

To achieve an extensive set of CBCT–facial scan pairs, it was necessary to imply strict criteria. To avoid any deliberate differences between the CBCT scan and facial scan:Facial expression: only calm, neutral faces with closed mouth and no facial expression were compared;Change in BMI: only pairs of scans separated by less than 7 days were included;Extreme artifacts: only CBCTs without extensive artifacts were included.

None of the above procedures (CBCT or 3D facial scanning) were performed exclusively for this study; rather, they were part of the diagnostic procedures during orthodontic planning. All included patients signed an informed consent form.

In the first part of our study, we collected 60 CBCT scans of patients (41 women and 19 men) and 60 3D facial scans acquired with the Bellus 3D Dental Pro application using the scanner. Bellus 3D Dental Pro is an application available for iPads and Apple’s iPhone 12 Pro. The patient’s face was scanned from every angle using the application’s instructions. The instructions were clear and easy to follow.

For each scan, patients had to maintain a neutral facial expression and relax both the jaw and eyes into a comfortable resting position. The measurement and display time is less than 10 s while the user rotates the head from left to right, allowing accurate 3D facial models to be created. This application helps capture a person’s face and create a high-resolution STL (Standard Tessellation Language) file, which is a 3D model of that object that can then be loaded into three-dimensional analysis software [25,36,37].

Both types of scans were taken at approximately the same time so that there would be no bias in the results due to weight gain or loss or other physical changes in the patient’s face that might occur over time.

### 2.2. Modification, Positioning and Analyzing of the Scans

The second part of our research was devoted to modifying and positioning the two scans in the correct positions so that they overlapped and could later be used to analyze specific facial areas, points, and attributes. This process consisted of several steps (Figure 2).

In this work, sixty TrueDepth scans of the face were compared with sixty corresponding facial surfaces segmented with CBCT. For the CBCT segmentation, the CBCT scan was segmented into Invivo 6 (Anatomage Inc., San Jose, CA, USA) in the Medical Design Studio module. Various programs can be used for the segmentation of CT/CBCT medical data [38] We used −700 opacity (Isosurface) where we could clearly see all facial soft tissues. We modified each CBCT scan so we could create a virtual “face shell” without unnecessary internal head structures. We used the function Freehand Volume Sculpting and removed excessive parts of the face. We also changed the subsample in the panel isosurface to 1. The final scan was saved in the STL file format. This file was opened in the Meshmixer program (Meshmixer™ (Autodesk^®®^, Inc., San Rafael, CA, USA) to process the face scan. We used the Brush function to further modify and select the areas of interest. Then the Optimize and Boundaries functions were used. We used the Invert and Erase functions to obtain the shape of our final face shell. Then, we removed artifacts using the Analysis Inspector function. We then exported this final CBCT scan and saved it again in STL binary format. All 60 CBCT face scans were modified in this way.

We used the above modified final CBCT scans and 3D TrueDepth model scans of patients and opened them both in the In Vivo Dental program. We manually positioned them to overlap the best. Then, we used the Mesh registrations function so that the scans overlapped even more. In this function, we used points that change less for different facial expressions, such as forehead, temples, and nose. We then exported these positioned TrueDepth model scans in STL binary format.

In the next step, we opened our final CBCT “face shell” model scan and our positioned TrueDepth model scan in the Meshmixer™ (Autodesk^®®^, Inc., San Rafael, CA, USA) program. We used the Analysis—Deviation function with 3 mm-maximum deviation. This function highlighted specific deviations between the two different scans.

### 2.3. Data Analysis and Comparison

In the third part of the research, we created a chart with different facial areas, points, and attributes. We measured 21 facial locations and their deviations of CBCT to TrueDepth scans (Figure 3). We recorded if the deviation was present on the specific facial location of each patient. We specifically focused on the deviations of Aesthetic and Harmony lines of these scans. We decided the clinical evaluation of the amount of deviation of TrueDepth scans to CBCT scans and afterwards conclude any protentional usage of the TrueDepth scanner in dental medicine (Table 1). We suggested that 3 mm or more of deviation should be clinically relevant and are questioning the suitability of this method.

It is known that even a subtle facial expression may cause significant volumetric changes in the face [39]. As any facial expressions identified on CBCT or facial scans were excluded from the sample, only minimal unrecognizable differences slipped the attention of evaluators of estimated ranges up to two millimeters. It is also known that the skin of the face undergoes circadian rhythm changes in a pattern of volumetric changes throughout the day and is also dependent on the water intake and body metabolism with changes that typically does not exceed 2 mm [40,41,42,43]. 

As the process of superimposition can be inaccurate, two independent operators participated in superimposition procedures of the 3D mesh—pairs with utilization of the automated best fitting algorithm. A 100% agreement was reached between both operators in the evaluation of matching discrepancies between scans, which eliminated the need for another operator or calibration of the evaluators. Alignment surfaces of the meshes included for the best-fit algorithm were predominantly the forehead area and other large surfaces of the face, including the cheekbone area that does not suffer from CBCT artifacts of marginal areas of CBCT, as well as artifacts caused by metallic dental fillings or prosthetic works. The CBCT scan was considered as reference, similar to a study published by Revilla-León et al. in 2021 [44]. In this study, a difference between CBCT and Face Camera Pro Bellus was evaluated, and it also confirmed a non-normal distribution of trueness and precision values (*p* < 0.05). 

Following these findings and considerations, a professional clinical orthodontic consideration was made to define differences greater than 3 mm as clinically relevant to compensate for potential bias in the range less than 3 mm that might result from subtle changes in facial expression or variations in skin volume during circadian cycles.

Differences between the aligned 3D meshes were visualized as heatmaps that disregarded positive or negative overlaps in favor to the absolute difference between the mesh surfaces. This absolute difference also does not reference any particular cephalometric points in (x, y, z) directions.

To compare the results in women and men, we used Fisher’s exact test in the contingency tables because the expected frequencies were low in most cases. All tests were performed at a significance level of 0.05.

To compare the scans from the lateral view, we selected seven different sites: the frontal region, nasal tip, philtrum, vermillion border, oral fissure, mentolabial sulcus, and mental region. To compare these seven sites, we used the Friedman test. Since the Friedman test is not parametric, it uses ranks. The higher the value, the higher the rank. 

The statistics were analyzed in the statistical software IBM SPSS 21.

To compare seven different locations in the middle of the face, we used Friedman test. Since the Friedman test is non-parametric, it works with ranks. The higher the value, the higher the rank. 

To compare results in women in men, we used Fisher’s exact test in the contingency tables because in most cases, the expected frequencies were low. All tests were made at an alpha significance level equal to 0.05.

## 3. Results

The amount of clinically relevant deviation (differences greater than 3 mm) was measured between the TrueDepth (Bellus3D Dental Pro) scan and the CBCT scan at specific locations on the face (Table 2). The lowest total deviation (less than or equal to 10%) was observed at the tip of the nose, bridge of the nose, nasolabial sulcus, philtrum, mentolabial sulcus, zygomatic bone, infraorbital region, and cheek region. A higher number of deviations (more than 30%) was observed on the right ala nasi, oral fissure, temporal region, and orbital region, with the highest values in the orbital region (65 and 68.3%).

Some differences were found between the right and left parts of some paired regions, such as in the ala nasi and the labial commissure. Significant differences in the amounts of deviations were also observed in males compared with females, statistically confirmed by Fisher’s exact 2-sided test in the frontal and right orbital regions. The amounts of clinically relevant deviations of all measured facial regions between the two scans in men, women, and overall are indicated by colored difference maps in Figure 3a,b. As can be seen in the figure, the deviations are lowest in the mid and lower facial regions, with higher accuracy in the facial prominences and lower accuracy in the deeper structures.

In addition, a double-check comparison was performed in seven regions visible from the lateral view of both aligned facial shells using Friedmann’s test. Since the Friedmann test is non-parametric, it works with ranks. The highest average rank was for oral fissures (4.94), the lowest rank for the sulcus mentolabialis (2.8) (Table 3).

Figure 4a shows the percentage of deviation in TrueDepth scans from CBCT scans in men and Figure 4b in women. Figure 5 shows the complete results for both.

## 4. Discussion

Our results confirm that scans created with TrueDepth sensors (Bellus3D Dental Pro App) are more accurate than one would think at first glance. This study states that the prominences of the face are more accurately imaged, but the accuracy of the concave structures is significantly worse, with higher accuracy in the middle and lower regions of the face. On the other hand, the deviation of 3 mm is still a significant difference from the point of view of orthodontic treatment in terms of micro-aesthetics, but not for a general area of the face. The era of facially driven orthodontics enabled by affordable, noninvasive facial scanning technologies will begin alongside increased capabilities of ubiquitous facial scanners in the form of cell phones and tablets. For macro-proportional facial assessment, the technology is already mature. However, for micro-aesthetic assessment, particularly around the lips, it is not mature yet.

It can be said that the cell phone scanning approach cannot yet compete with CBCT scans in terms of accuracy, although it clearly dominates in terms of availability and noninvasiveness. Higher accuracy in CBCT scans improves the quality of the data recorded by the scanner, which ultimately improves the outcome. The current accuracy of the TrueDepth sensor used with the Bellus3D Dental Pro application is not sufficient to achieve high facial accuracy and therefore cannot be used for detailed orthodontic treatment planning.

CBCT scans, which were used as the gold standard in this study, are not the ideal imaging modality for 3D facial soft tissue evaluations. Primarily, stereophotogrammetry should have been considered for evaluating the accuracy of the captured facial image from the TrueDepth technology of the smartphone. Possible errors in heatmap regional evaluation were reduced with assessment by two independent evaluators that were in agreement on all evaluated pairs, as well as repeated alignment by the described method, which never resulted in significantly different alignments by any factor.

The reproducibility of the captured 3D Images using smartphones with Bellus3D Pro was reasonable. With proper posture as a strict limitation of the face-scanning procedure, the resulting facial scans had submillimeter discrepancies. Poor repeatability of facial scanning may impact negatively on the validity of the method. 

In a recent study of D’Ettorre et al. in 2022 [45], the surface-to-surface deviation analysis between Bellus3D and 3dMD(stereophotogrammetry) showed an overlap percentage of 80.01% ± 5.92% within the range of 1 mm discrepancy. A recent systematic review paper focusing on stereophotogrammetry and smartphone technology by Quinzi et al., 2022, concluded that “Stationary stereophotogrammetry devices showed a mean accuracy that ranged from 0.087 to 0.860 mm, portable stereophotogrammetry scanners from 0.150 to 0.849 mm, and smartphones from 0.460 to 1.400 mm.” [46]. The volumetric estimation errors are typically bigger in smartphone scanning than in photogrammetry [47].

A limitation of this study is the small initial number of participants, where the ratio of women and men was not balanced. This could cause unexpected differences between men and women or between the right and left sides of some paired facial regions.

Another limitation of this study is the comparison of the temporal and frontal regions between the CBCT and the TrueDepth scans, which suffered from major artifacts in these regions. The upper border of the CBCT was located typically in the middle horizontal level of the frontal bone, representing many artifacts. Despite the crop of these artifacts, the data of the upper part of the forehead often contained artifacts or were missing in the CBCT scans (Figure 6, left). In consideration of why the discrepancy was observed most frequently in the orbital region, the probable explanation is differences between closed and opened eyelids. Closed eyes in CBCT scans were typical (Figure 6b), whereas open eyes in the TrueDepth scan could influence the result (Figure 6, right scan).

Sensors are frequently utilized in orthodontics [48,49]. According to the recent study of Cho et al., 2022, “From the 3D CT images, 3D models, also called digital impressions, can be computed for CAD/CAM-based fabrication of dental restorations or orthodontic devices. However, the cone-beam angle-dependent artifacts, mostly caused by the incompleteness of the projection data acquired in the circular cone-beam scan geometry, can induce significant errors in the 3D models.” [50]. Even research of Pojda et al., 2021, confirms the necessity of cheaper and more convenient alternatives to orthodontic facial 3D imaging [51]. Sensors for optical scanning and high-definition CT scanning (microCT) are frequently utilized on the borders of orthodontics and forensic dentistry in the identification of dental patterns, opening new possibilities of human remains identification [52].

A key context to this discussion is that the TrueDepth scanner available on an Apple smartphone or tablet has an advantage in terms of cost and availability, the scans can be performed in a short time with real-time processing, and, of course, the scans can be performed without the patient receiving radiation. In addition, it could open new opportunities for telemedicine applications in dental practices.

Moreover, the TrueDepth scanner, which may be used on certain Apple devices, has tremendous potential, and could potentially be used in a variety of medical settings in the future. Given the pace of development of smartphones and applications, we can predict that the precision and quality of scans will gradually improve.

## 5. Conclusions

The results confirmed statistically significant differences between facial surfaces from CBCT and TrueDepth (Bellus3D Dental Pro cell phone application) scans in some facial regions with amplitudes greater than 3 mm. This suggests that the current TrueDepth sensor from Apple has limited clinical use in orthodontic applications. However, clinical application of the described approach in orthodontic facial analysis, which does not require accuracy within 3 mm, can be considered.

## Figures and Tables

**Figure 1 sensors-22-07752-f001:**
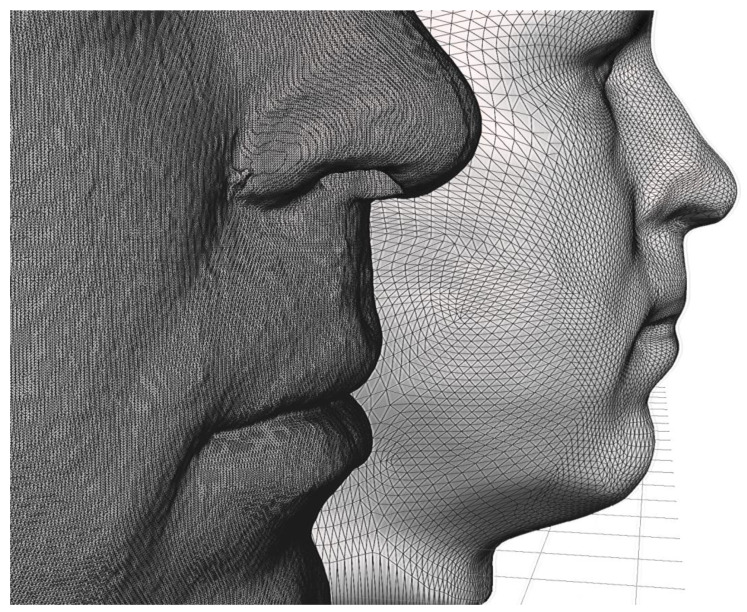
The difference between the polygon resolution of the STL from the CBCT (in the front) and the STL from the face scan (TrueDepth—Bellus3D Dental Pro) in the back is noticeable.

**Figure 2 sensors-22-07752-f002:**
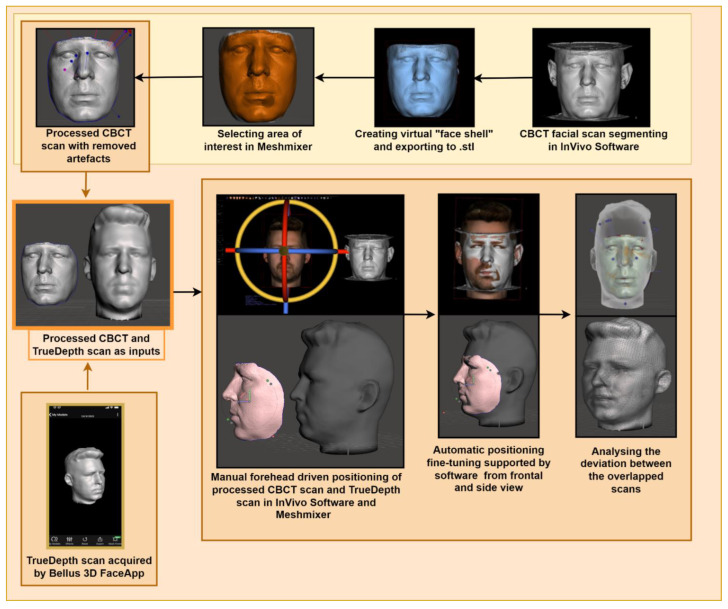
The schematic approach of the proposed method.

**Figure 3 sensors-22-07752-f003:**
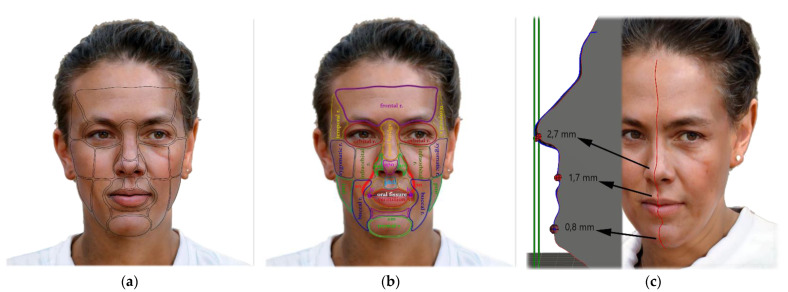
AI face generator was used to create this hermaphrodite face. It shows: (**a**) Observed facial locations; (**b**) Abbreviations of locations used: region (r.), parotid-masseteric region (pmr), labials commissure (lc), ala nasi (an), apex of nose (apn), sulcus nasolabialis (sn), sulcus mentolabialis (sm), vermillion border (vb), philtrum (phil); (**c**) Example of comparison of sagittal sections of the facial profile of the CBCT surface shell and the Bellus 3D shell. In this way, the differences between the shells were measured and the results of the differential maps were checked in the median plane.

**Figure 4 sensors-22-07752-f004:**
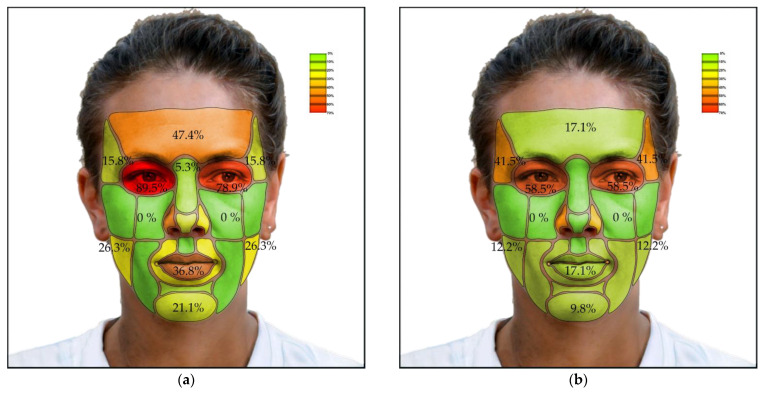
The percentage of deviation in TrueDepth scans from CBCT scans: (**a**) Results in men; (**b**) Results in women.

**Figure 5 sensors-22-07752-f005:**
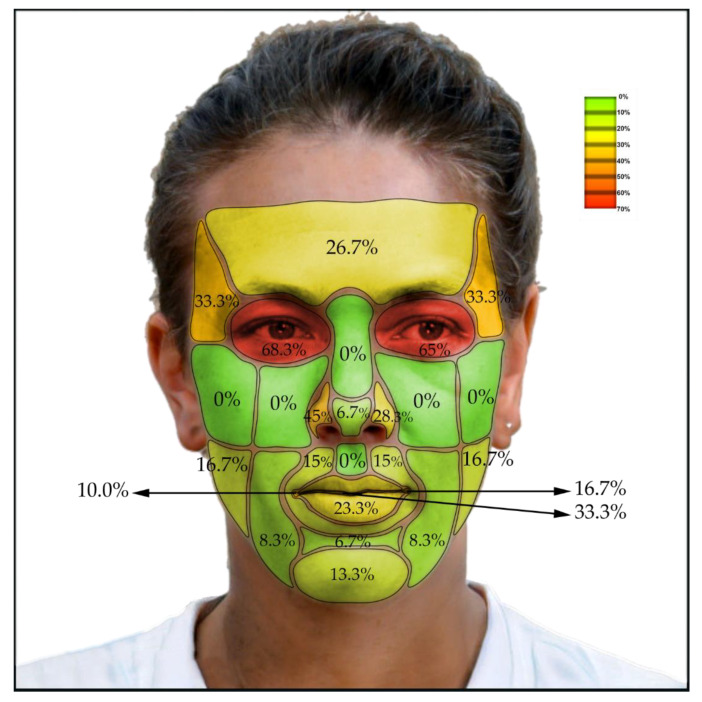
The percentage of deviation in TrueDepth (Bellus3D Dental Pro) scans from CBCT scans—all results together.

**Figure 6 sensors-22-07752-f006:**
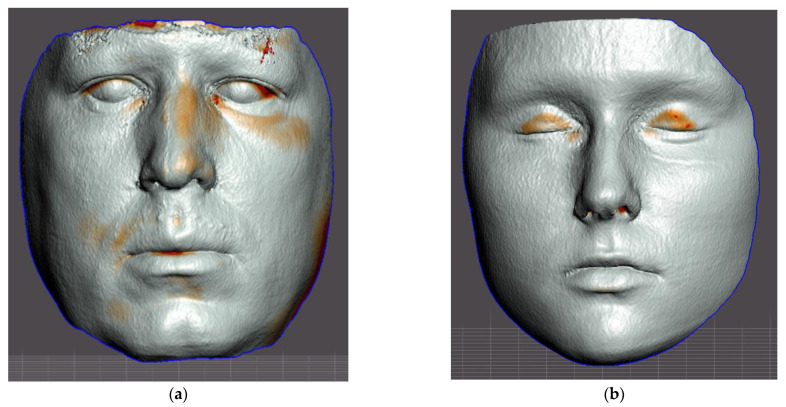
Examples of aligned segmented CBCT face surfaces and face TrueDepth–Bellus3D scans of authors of this paper. On the resulting differential maps (heatmaps), red indicates differences between both scans: (**a**) Left surface shows frequent disruptive artifacts on the upper border of CBCT scan; (**b**) The right scan shows frequent differences resulting from eyes closed during the CBCT scanning.

**Table 1 sensors-22-07752-t001:** Clinical evaluation of amount of deviation between CBCT and 3D TrueDepth (Bellus3D Dental Pro) scans.

Intervals of Deviation (mm)	Approximation	Clinical Explanation
0	0	clinically irrelevant
0–1	0.5	clinically irrelevant
1–3	2	clinically relevant only in extreme detail evaluations for micro-aesthetics
3+	4	clinically relevant, questioning suitability of this method

**Table 2 sensors-22-07752-t002:** The amount of deviation of TrueDepth (Bellus3D Dental Pro) scans from CBCT scans. First column represents the facial location; second the percentage of men with deviation more than 3 mm between these scans; third column, the same as the second but in women; and the last column represents the difference between men and women evaluated using Fisher’s exact test. In the last column, bold values represent significant differences.

Facial Location	>3mm Men	>3mm Women	>3mm Total	Fisher’s Exact Test (2-Sided)
apex of nose	15.8%	2.4%	6.7%	0.116
ala nasi L	26.3%	29.3%	28.3%	1.000
ala nasi R	42.1%	46.3%	45.0%	0.788
nasal bridge L	5.3%	4.9%	5.0%	0.797
nasal bridge R	5.3%	4.9%	5.0%	0.797
sulcus nasolabialis L	0	2.4%	1.7%	0.850
sulcus nasolabialis R	0	4.9%	3.3%	1.000
philtrum	0	0	0	0.365
vermilion border	26.3%	9.8%	15.0%	0.171
vermilion	36.8%	17.1%	23.3%	0.182
oral/labial commisure L	10.5%	19.5%	16.7%	0.453
oral/labial commisure R	10.5%	9.8%	10.0%	0.147
oral fissure	26.3%	36.6%	33.3%	0.342
sulcus mentolabialis	5.3%	7.3%	6.7%	0.908
zygomatic region	0	2.4%	1.7%	0.243
temporal region	15.8%	41.5%	33.3%	0.140
frontal region	47.4%	17.1%	26.7%	**0.047**
orbital region L	78.9%	58.5%	65.0%	0.154
orbital region R	89.5%	58.5%	68.3%	**0.019**
infraorbital region	0	4.9%	3.3%	1.000
mental region	21.1%	9.8%	13.3%	0.601
parotid-masseteric region	26.3%	12.2%	16.7%	0.406
buccal region	0	12.2%	8.3%	0.184

**Table 3 sensors-22-07752-t003:** The comparison of seven locations visible from side view of overlapped scans using Friedmann’s test. The locations are arranged from lowest to highest rank.

Location	Rank
sulcus mentolabialis	2.80
philtrum	3.08
apex of nose	4.01
mental region	4.33
Vermillion border	4.34
frontal region	4.50
oral fissure	4.94

## Data Availability

Anonymized data supporting the reported results are freely available at: https://docs.google.com/spreadsheets/d/1wfbPd66RdPE4EIjg6c4jTTp5-kJH5dBWIPBwEODouMw/edit?usp=sharing (accessed on 12 September 2022). The authors ensure that the data shared are in accordance with the consent provided by participants on the use of confidential data.

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
