# Peer review of "Smartphone-Based Facial Scanning as a Viable Tool for Facially Driven Orthodontics?"

_sensors, 2022, doi:10.3390/s22207752_

Round 1

Reviewer 1 Report

The idea of the article is very actual and interesting. However, the authors must clarify:

-how  was determined the lot number

-page 6 raw 197- "All 50 CBCT face scans were modified in this way"- isn't the initial number of CBCT 60? 

-page 6 raw - "We suggested that 3 and more millimeters of deviation should be clinically relevant and are questioning the suitability of this method."-I suggest that the authors document this affirmation very well because the interpretation of all results depends on it. 

-the process of superimposing surfaces can be inaccurate- how many tests and measurements have been done and how many investigators were involved? 

- I suggest to consider this study: Revilla-León M, Zandinejad A, Nair MK, Barmak BA, Feilzer AJ, Özcan M. Accuracy of a patient 3-dimensional virtual representation obtained from the superimposition of facial and intraoral scans guided by extraoral and intraoral scan body systems. J Prosthet Dent. 2021 Apr 7:S0022-3913(21)00106-2. doi: 10.1016/j.prosdent.2021.02.023. Epub ahead of print. PMID: 33838919.

Author Response

Dear reviewer,

We value your positive feedback. We have revised the paper in accordance with your recommendations. Below we add responses to each of your remarks.

The idea of the article is very actual and interesting. However, the authors must clarify:

-how  was determined the lot number

This was calculated and suggested by a professional statistician listed in acknowledgements.

-page 6 raw 197- "All 50 CBCT face scans were modified in this way"- isn't the initial number of CBCT 60? 

Thank you for noticing, that was a slip.

-page 6 raw - "We suggested that 3 and more millimeters of deviation should be clinically relevant and are questioning the suitability of this method."-I suggest that the authors document this affirmation very well because the interpretation of all results depends on it. 

Thank you for this suggestion. We have elaborated on our proposition on lines 217+ Adding the following text and 5 new references:

“It is known that even a subtle facial expression may cause significant volumetric change in the face [38]. As any facial expressions identified on CBCT or facial scans were excluded from the sample only minimal unrecognizable differences slipped the attention of evaluators of estimated ranges up to two millimeters. It is also known that skin of the face undergoes circadian rhythm changes in a pattern of volumetric changes throughout the day and is also dependent on the water intake and body metabolism with changes that typically does not exceed 2mm [39–42].

-the process of superimposing surfaces can be inaccurate- how many tests and measurements have been done and how many investigators were involved? 

- I suggest to consider this study: Revilla-León M, Zandinejad A, Nair MK, Barmak BA, Feilzer AJ, Özcan M. Accuracy of a patient 3-dimensional virtual representation obtained from the superimposition of facial and intraoral scans guided by extraoral and intraoral scan body systems. J Prosthet Dent. 2021 Apr 7:S0022-3913(21)00106-2. doi: 10.1016/j.prosdent.2021.02.023. Epub ahead of print. PMID: 33838919.

We have addressed your remark above and added text in the Methods section with better explanation of the method and evaluation. We have also considered your suggested reference, and added five more explaining common volumetric fluctuations of the face and effect of mild facial expressions. We have added the following text and references on lines 247-258:

“As the process of superimposition can be inaccurate, two independent operators participated in superimposition procedures of the 3D mesh - pairs with utilization of automated best fitting algorithm. 100% agreement between both operators was reached in evaluation of matching discrepancies between scans, which eliminated the need for another operator or calibration of the evaluators. CBCT scan was considered as reference similar to study published by Revilla-León et al. 2021 [43]. In this study a difference between CBCT and Face Camera Pro Bellus was evaluated, and it also confirmed a not normal distribution of trueness and precision values (P<.05).

Following these findings and considerations, a professional clinical orthodontic consideration was made to define differences greater than 3 mm as clinically relevant to compensate for potential bias in the range less than 3 mm that might result from subtle changes in facial expression or variations in skin volume during circadian cycles.

  1. Rawlani, R.; Qureshi, H.; Rawlani, V.; Turin, S.Y.; Mustoe, T.A. Volumetric Changes of the Mid and Lower Face with Animation and the Standardization of Three-Dimensional Facial Imaging. Plast Reconstr Surg 2019, 143, 76–85, doi:10.1097/PRS.0000000000005082.
  2. le Fur, I.; Reinberg, A.; Lopez, S.; Morizot, F.; Mechkouri, M.; Tschachler, E. Analysis of Circadian and Ultradian Rhythms of Skin Surface Properties of Face and Forearm of Healthy Women. Journal of Investigative Dermatology 2001, 117, 718–724, doi:10.1046/J.0022-202X.2001.01433.X.
  3. Reinberg, A.; Koulbanis, C.; Soudant, E.; Nicolai, A.; Mechkouri, M.; Smolensky, M. Day-Night Differences in Effects of Cosmetic Treatments on Facial Skin. Effects on Facial Skin Appearance. http://dx.doi.org/10.3109/07420529009056956 2009, 7, 69–79, doi:10.3109/07420529009056956.
  4. Ferrario, V.F.; Sforza, C.; Poggio, C.E.; Schmitz, J.H. Facial Volume Changes During Normal Human Growth and Development. Anat. Rec 1998, 250, 480–487, doi:10.1002/(SICI)1097-0185(199804)250:4.
  5. Mailey, B.; Baker, J.L.; Hosseini, A.; Collins, J.; Suliman, A.; Wallace, A.M.; Cohen, S.R. Evaluation of Facial Volume Changes after Rejuvenation Surgery Using a 3-Dimensional Camera. Aesthet Surg J 2016, 36, 379–387, doi:10.1093/ASJ/SJV226.
  6. Revilla-León, M.; Zandinejad, A.; Nair, M.K.; Barmak, B.A.; Feilzer, A.J.; Özcan, M. Accuracy of a Patient 3-Dimensional Virtual Representation Obtained from the Superimposition of Facial and Intraoral Scans Guided by Extraoral and Intraoral Scan Body Systems. Journal of Prosthetic Dentistry 2021, 0, doi:10.1016/j.prosdent.2021.02.023.

We believe we have addressed all your recommendations and remarks.

Thank you for your guidance.

Kind regards

authors

Reviewer 2 Report

Thank you for asking me to review this article, the authors to be congratulated for this interesting piece of research, the following are my concerns:

1. CBCT scans which were used as the gold standard in this study are not the ideal imaging modality of the 3D soft tissue of the face. Stereophotogrammetry should have been considered for the evaluation of the accuracy of the record facial image suing the TruDepth technology of the smart phone. This, at least,  should be discussed. 

2. The authors used Mesh registration method to superimpose the corresponding images without explaining which anatomical points have initiated this process.  

3. The errors of the methods including the repeated landmarking and the superimposition of the corresponding images were not provided. 

4.  The authors should comment on the reproducibility of the captured 3D images using smartphones, this may impact negatively on the validity of the methods

5. No justification to the 3 mm cut off for the accuracy of the recorded images using smart phone.

6. What was the reason of the maximum discrepancies of the captured images at deep facial curvatures? what was the reason for the differences between the level of accuracy between the right and the left side of the nose?

7. It is not clear from the analysis if the direction (x, y, z) of the differences between the two images was taken in consideration, this needs further clarification.

Author Response

Thank you for asking me to review this article, the authors to be congratulated for this interesting piece of research, the following are my concerns:

Dear reviewer,

We appreciate your positive feedback. We have revised the paper in accordance with your recommendations. Below we include responses to each of your comments.

  1. CBCT scans which were used as the gold standard in this study are not the ideal imaging modality of the 3D soft tissue of the face. Stereophotogrammetry should have been considered for the evaluation of the accuracy of the record facial image suing the TruDepth technology of the smart phone. This, at least,  should be discussed. 

Thank you for your remark, we agree. We have added this point into Discussion on lines 337-341 as the following text:

“CBCT scans, which were used as the gold standard in this study, are not the ideal imaging modality for 3D facial soft tissue evaluations. Primarily stereophotogrammetry should have been considered for evaluating the accuracy of the captured facial image using the TrueDepth technology of the smartphone.

  1. The authors used Mesh registration method to superimpose the corresponding images without explaining which anatomical points have initiated this process.  

Thank you for this point, we have elaborated on the process of mesh-alignment and added the following text in the Methods section on lines 240-262:

“As the process of superimposition can be inaccurate, two independent operators participated in superimposition procedures of the 3D mesh - pairs with utilization of automated best fitting algorithm. 100% agreement between both operators was reached in evaluation of matching discrepancies between scans, which eliminated the need for another operator or calibration of the evaluators. Alignment surfaces of the meshes included for the best fit algorithm were predominantly forehead area and other large surfaces of the face including cheekbone area that does not suffer from CBCT artefacts of marginal areas of CBCT as well as artefacts caused by metallic dental fillings or prosthetic works. CBCT scan was considered as reference similar to study published by Revilla-León et al. 2021 [43]. In this study a difference between CBCT and Face Camera Pro Bellus was evaluated, and it also confirmed a not normal distribution of trueness and precision values (P<.05).

Following these findings and considerations, a professional clinical orthodontic consideration was made to define differences greater than 3 mm as clinically relevant to compensate for potentional bias in the range less than 3 mm that might result from subtle changes in facial expression or variations in skin volume during circadian cycles.

  1. The errors of the methods including the repeated landmarking and the superimposition of the corresponding images were not provided. 

We have discussed the errors/limitations of the method including the superimposition. Landmarking was not performed, the heatmaps were evaluated by 2 independent observers which is now better explained in the Methods. We added the following text in the Methods section on lines 240-262, within the previous paragraph. Another text has been added to Discussion in lines 342-348:

“CBCT scans, which were used as the gold standard in this study, are not the ideal imaging modality for 3D facial soft tissue evaluations. Primarily stereophotogrammetry should have been considered for evaluating the accuracy of the captured facial image using the TrueDepth technology of the smartphone. Possible errors in heatmap regional evaluation were reduced with assessment by two independent evaluators that were in agreement in all evaluated pairs as well as repeated alignment by described method never resulted in significantly different alignment by any factor.

  1. The authors should comment on the reproducibility of the captured 3D images using smartphones, this may impact negatively on the validity of the methods

We have commented on the reproducibility of the performed facial scans as well as potential negative impacts in methods and Discussion in lines 321-339:

CBCT scans, which were used as the gold standard in this study, are not the ideal imaging modality for 3D facial soft tissue evaluations. Primarily stereophotogrammetry should have been considered for evaluating the accuracy of the captured facial image using the TrueDepth technology of the smartphone. Possible errors in heatmap regional evaluation were reduced with assessment by two independent evaluators that were in agreement in all evaluated pairs as well as repeated alignment by described method never resulted in significantly different alignment by any factor.

The reproducibility of the captured 3D images using smartphones using Bellus3D Pro was reasonable. With proper posture a strict limitation to face-scanning procedure the resulting facial scans had submillimeter discrepancies. Poor repeatability of facial scanning may impact negatively on the validity of the method.

In a recent study of D'Ettorre et al. 2022 [44], the surface-to-surface deviation analysis between the Bellus3D and 3dMD(stereophotogrammetry) showed an overlap percentage of 80.01% ± 5.92% within the ranges of 1 mm discrepancy. Recent systematic review paper focused on Stereophotogrammetry and Smartphone Technology by Quinzi et al. 2022 concluded that "Stationary stereophotogrammetry devices showed a mean accuracy that ranged from 0.087 to 0.860 mm, portable stereophotogrammetry scanners from 0.150 to 0.849 mm, and smartphones from 0.460 to 1.400 mm.” [45]. The volumetric estimation errors are typically bigger in smartphone scanning than in photogrammetry [46].”

  1. No justification to the 3 mm cut off for the accuracy of the recorded images using smart phone.

Thank you for pointing this out, we have elaborated on the reasons of 3 mm cutout and explained it better. We have added the following text in the Methods section on lines 240-262:

“As the process of superimposition can be inaccurate, two independent operators participated in superimposition procedures of the 3D mesh - pairs with utilization of automated best fitting algorithm. 100% agreement between both operators was reached in evaluation of matching discrepancies between scans, which eliminated the need for another operator or calibration of the evaluators. Alignment surfaces of the meshes included for the best fit algorithm were predominantly forehead area and other large surfaces of the face including cheekbone area that does not suffer from CBCT artefacts of marginal areas of CBCT as well as artefacts caused by metallic dental fillings or prosthetic works. CBCT scan was considered as reference similar to study published by Revilla-León et al. 2021 [43]. In this study a difference between CBCT and Face Camera Pro Bellus was evaluated, and it also confirmed a not normal distribution of trueness and precision values (P<.05).

Following these findings and considerations, a professional clinical orthodontic consideration was made to define differences greater than 3 mm as clinically relevant to compensate for potentional bias in the range less than 3 mm that might result from subtle changes in facial expression or variations in skin volume during circadian cycles.

  1. What was the reason of the maximum discrepancies of the captured images at deep facial curvatures? what was the reason for the differences between the level of accuracy between the right and the left side of the nose?

Thank you for noticing. We have been concerned about this during the research. We preferred not to speculate directly in the paper, albeit simple explanation for the first question is that the Bellus Dental Pro APP is using some kind of undocumented transfer mask to create more primitive mesh suitable for 3D model export. This primitivized mesh loses most of the extreme concavities as well as some unexpected convexities in favor to respecting rather larger areas than smaller structures.. Also, when 3D face scan was exported all deeper and smaller details around nose and lips seem to be sheathed. In regard to surprising difference between left and right side of ala nasi we have checked all the sourcing data and we have learned that it was not caused by any extreme result, rather it was connected to previously described problem, that 3D mesh reconstruction in this app did not favor small structures like area over major alar cartilage and was perceiving deformation in this zones frequently. In conclusion, the difference is caused by coincidence that this primitivized registration was not balanced.

  1. It is not clear from the analysis if the direction (x, y, z) of the differences between the two images was taken in consideration, this needs further clarification.

We have extended the Methods section for this description in lines 242-245 with the new text:

“Differences between the aligned 3D meshes were visualized as heatmaps which disregarded positive or negative overlap in favor to absolute difference between mesh surfaces. This absolute difference also does not referencing any particular cephalometric points in (x, y, z) directions.

Dear reviewer, we believe we have addressed all your recommendations and remarks.

Thank you for your guidance.

Kind regards

authors

Reviewer 3 Report

Authors show in a study on 60 persons that the smartphones` TrueDepth sensors are not suited to analyze facial fotos for orthodontics questions, as changes below 3 mm cannot be resolved.

Study is nicely done and presented, still there is little to learn from this negative result. After reading the paper it was not clear what os the take home message, apart fm the to be expected fact that spcific hard and software is better suited for orthodontics questions than smartphone based one.

Also other, more developped facial recognistion software, like Face To Gene Software was not mentioned in the introduction.

Author Response

Authors show in a study on 60 persons that the smartphones` TrueDepth sensors are not suited to analyze facial fotos for orthodontics questions, as changes below 3 mm cannot be resolved.

Study is nicely done and presented, still there is little to learn from this negative result. After reading the paper it was not clear what os the take home message, apart fm the to be expected fact that spcific hard and software is better suited for orthodontics questions than smartphone based one.

Also other, more developed facial recognistion software, like Face To Gene Software was not mentioned in the introduction.

Dear reviewer,

We appreciate your feedback. We have revised the paper in accordance to highlight the outcomes. We have added a text-paragraph to Introduction referencing the Face2 Gene Software with two extra references:

“Mobile phone 3D facial scanning in combination with AI algorithms incorporated in a smartphone app for example - Face2Gene (FDNA Inc., Boston MA, USA) are currently forming a powerful tool for early diagnostics. Diseases not only manifest as internal structural and functional abnormalities, but also have facial characteristics and appearance deformities. Specific facial phenotypes are potential diagnostic markers, especially for endocrine and metabolic syndromes, genetic disorders, facial neuromuscular diseases [35,36].”

we have also elaborated on the process of mesh-alignment and added the following text in the Methods section on lines 240-262:

“As the process of superimposition can be inaccurate, two independent operators participated in superimposition procedures of the 3D mesh - pairs with utilization of automated best fitting algorithm. 100% agreement between both operators was reached in evaluation of matching discrepancies between scans, which eliminated the need for another operator or calibration of the evaluators. Alignment surfaces of the meshes included for the best fit algorithm were predominantly forehead area and other large surfaces of the face including cheekbone area that does not suffer from CBCT artefacts of marginal areas of CBCT as well as artefacts caused by metallic dental fillings or prosthetic works. CBCT scan was considered as reference similar to study published by Revilla-León et al. 2021 [43]. In this study a difference between CBCT and Face Camera Pro Bellus was evaluated, and it also confirmed a not normal distribution of trueness and precision values (P<.05).

Following these findings and considerations, a professional clinical orthodontic consideration was made to define differences greater than 3 mm as clinically relevant to compensate for potentional bias in the range less than 3 mm that might result from subtle changes in facial expression or variations in skin volume during circadian cycles.

we have also rewritten the reasons of 3 mm cutout and explained it better as well as conclusion. We have added the following text in the Methods section on lines 240-262:

“As the process of superimposition can be inaccurate, two independent operators participated in superimposition procedures of the 3D mesh - pairs with utilization of automated best fitting algorithm. 100% agreement between both operators was reached in evaluation of matching discrepancies between scans, which eliminated the need for another operator or calibration of the evaluators. Alignment surfaces of the meshes included for the best fit algorithm were predominantly forehead area and other large surfaces of the face including cheekbone area that does not suffer from CBCT artefacts of marginal areas of CBCT as well as artefacts caused by metallic dental fillings or prosthetic works. CBCT scan was considered as reference similar to study published by Revilla-León et al. 2021 [43]. In this study a difference between CBCT and Face Camera Pro Bellus was evaluated, and it also confirmed a not normal distribution of trueness and precision values (P<.05).

Following these findings and considerations, a professional clinical orthodontic consideration was made to define differences greater than 3 mm as clinically relevant to compensate for potentional bias in the range less than 3 mm that might result from subtle changes in facial expression or variations in skin volume during circadian cycles.

Dear reviewer, we believe we have addressed all your recommendations and remarks.

Thank you for your guidance.

Kind regards

authors

Round 2

Reviewer 1 Report

I agree to be published in the present form

Reviewer 3 Report

All points of my previous review were adressed - thanks.